# Recent Progress on Anti-Slip and Highly Wear-Resistant Elastic Coatings: An Overview

**Wenrui Chen** [1,2,*,†] [ID], **Jingying Zhang** [1,2,†], **Xinyu Qi** [1,2], **Pan Tian** [1,2], **Zenghui Feng** [1,2], **Weihua Qin** [1,2], **Dongheng Wu** [1,2], **Lanxuan Liu** [1,2] and **Yang Wang** [1,2,*]

1   Wuhan Research Institute of Materials Protection, Wuhan 430030, China; zhangjy288@163.com (J.Z.);
    qixinyu3265196@126.com (X.Q.); tianpcam@163.com (P.T.); 13419694843@163.com (Z.F.);
    qinweihua0@163.com (W.Q.); 18804050206@163.com (D.W.); liulanxuan@rimp.com.cn (L.L.)
2   State Key Laboratory of Special Surface Protection Materials and Application Technology,
    Wuhan 430030, China
*   Correspondence: chenwenrui@rimp.com.cn (W.C.); wangyang@rimp.com.cn (Y.W.)
†   These authors contribute equally to this work.

**Abstract:** There has been great interest in the research and development of different anti-skid and highly wear-resistant materials that can effectively reduce energy losses and improve efficiency in numerous applications. This article reviews the design, performance, and application of anti-skid and highly wear-resistant coating materials at home and abroad. First, it introduces the structure and mechanism of anti-skid and wear-resistant coatings. The preparation of different anti-slip coatings is mainly accomplished by changing the base material and anti-slip granules as well as the coating method, and the anti-slip performance is determined with the coefficient of friction test. The application mostly encompasses airplane and ocean decks, as well as pedestrian spaces. This review introduces the development status and research progress of metal-based anti-skid coatings and polymer-based anti-skid coatings, which are two groups of pavement. Finally, the challenges and future development directions of this key field are summarized and considered.

**Keywords:** anti-slip; wear-resistant; elastic coatings; polymer



## 1. Introduction

Paint is a convenient and effective treatment option for enhancing anti-slip properties and wear resistance. It has the ability to be applied to various surface conditions and may also impart varied colors to items, enhancing their aesthetic appeal and serving as a decorative element [1]. It efficiently safeguards the underlying layer from erosion caused by temperature, water, salt, and several other detrimental elements [2,3]. Additionally, it significantly contributes to preventing dust accumulation and enhancing slip resistance [4,5].

Wear-resistant and anti-skid coatings are crucial in various fields. They are mostly utilized in locations that necessitate anti-skid properties, such as deck areas, including those found on maritime vessels and aircraft. Additionally, they are well-suited for heavy-duty vehicles operating on deck surfaces. Ensuring secure traction while driving, safe navigation through passageways, drilling platforms, and other related scenarios. The civil applications of wear-resistant and anti-skid coatings primarily involve the surfaces of large equipment in humid environments, such as ships in marine environments [6–8]. They are also used in pedestrian walkways, sports venues, parking lots, bathrooms, swimming pools, and other areas where anti-skid properties are required [9,10]. Additionally, wear-resistant and anti-slip paper is used between product packages [11–14].

The safety of personnel and equipment is directly influenced by the effectiveness of anti-skid and highly wear-resistant coatings [15–18]. Based on market analysis from Markets and Markets, the worldwide market value of anti-skid coatings was USD 105 million in 2018 and is projected to reach USD 161 million by 2023 [19].

During the initial stages of the development of China, cement and yellow sand were frequently employed as primary components for the production of anti-skid and highly durable coatings. They exhibit deficiencies like as inadequate durability, limited wear resistance, and subpar cold and heat-shrinking properties. Simultaneously, they are susceptible to freezing and fracturing in low-temperature conditions. Further enhancements were developed by utilizing polyurethane, epoxy polyamide, and other resins or metals as foundational materials, and including emery, silicon carbide, and similar substances as anti-skid particles to create anti-skid coatings, resulting in a noticeable improvement in effectiveness [20–22].

## 2. Anti-Slip and Highly Wear-Resistant Elastic Coating

### 2.1. Anti-Slip Mechanism of Coating

The anti-slip mechanism of coatings is determined by the friction coefficient of the coating layer that is established upon curing. In general, a coating can be classified as an anti-slip coating when the friction coefficient of the cured surface exceeds 0.45 [21–23]. Hence, enhancing the friction coefficient and minimizing wear can enhance the anti-skid properties of the coating. An anti-slip coating is a type of coating that is applied to materials in order to enhance the level of friction on their surfaces. To enhance the friction coefficient of the coating surface, it is customary to incorporate irregularly shaped anti-skid particles into the coating. Upon completion of the curing process, a portion of the anti-skid particles will extend outward from the surface of the coating, resulting in a relief-like structure. This imparts a specific level of roughness to the coating surface, effectively preventing any potential sliding between the object and the substrate surface. As a result, the coating exhibits anti-slip capabilities.

Various coatings can be chosen based on specific uses and requirements. The types are also more varied. The types demonstrate a higher level of variety. The coating method can be classified as spraying, brushing, and roller coating, depending on the employed methodology. Spraying, dipping, and brushing are methods employed to generate practical coatings. During the dip coating process, the substrate is submerged into the functional ink formulation. An even coating can be achieved by precisely determining the withdrawal speed, outer viscosity, temperature, and other relevant external parameters prior to withdrawing the substrate. In addition to immersing the substrate in the functional ink, the ink can also be applied to the substrate by means of a spraying cannon. The wet coating procedure is frequently employed to produce coverings that encompass expansive surfaces. Additionally, the functional ink can be applied to a substrate using a painting brush.

With appropriate usage and gradual deterioration, the anti-skid coating will experience an enhancement in its anti-skid capabilities. The presence of anti-skid granules is attributed to their dispersion within the matrix and their partial solidification on the surface. Upon wearing, the uncovered edges and corners of the granules generate heightened resistance, hence leading to an increase in the friction coefficient [24–26]. Anti-skid particles can be incorporated into the coating using several methods, such as direct addition, hollow particle crushing, the closed layer approach, or mixing [27], as seen in Figure 1A (1 is base materials, 2 is base coatings, 3 and 10 are protective film, 4–8 are anti-skid particles, 9 is soot particles). The researchers from the University of Delaware, led by D.L. Burris et al. [28], have demonstrated that the incorporation of nanoparticles can significantly decrease coating wear and alter the surface morphology of the coating, as depicted in Figure 1B, and they thought that the primary role of the nanoparticles was the initiation of a unique wear resistant morphology in the polymer.

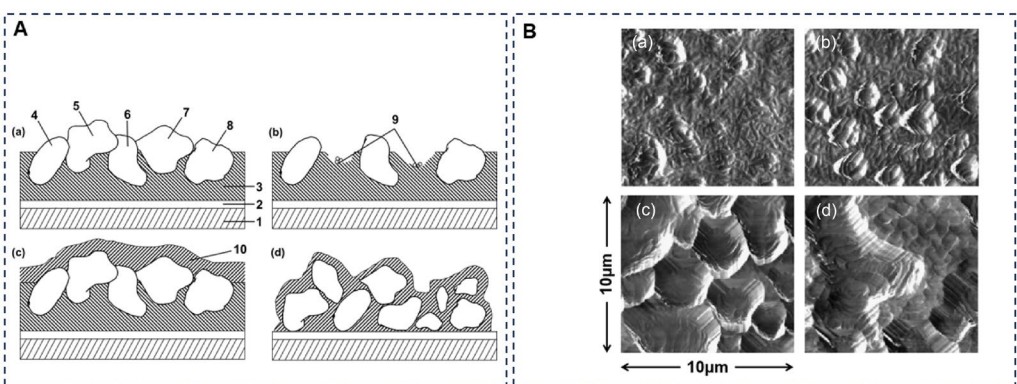

**Figure 1.** (**A**) Diagram illustrating the structure of the coating after the inclusion of anti-slip particles ((a) direct addition; (b) hollow particle crushing; (c) the closed layer approach; (d) mixing method) [27]; (**B**) atomic force microscopy (AFM) images showing the coating phase before and after the addition of nanoparticles ((a) neat PTFE; (b) 0.5 vol% 40 nm δ:γ phase alumina-PTFE; (c) 0.5 vol% 40 nm α phase alumina-PTFE treated with fluorosilane-treated nanoparticles; (d) 0.5 vol% 40 nm α phase alumina-PTFE with untreated nanoparticles) [28].

### 2.2. Wear-Resistant Mechanism of Coating

Material wear encompasses several wear mechanisms, including abrasion, fretting, adhesion, fatigue, oxidation, and other tribochemical reactions. The process of wear can be categorized into three distinct states: minimal, moderate, and severe wear. In the low wear state, the stress level is below the elastic limit and is typically marked by localized deformation known as micro-bulging. In the light wear state, the stress level is below the plastic limit, resulting in the presence of microcracks, local fractures, and small wear particles [29]. The stress level in the severe wear condition is approximately equal to or surpasses the critical failure stress [30], resulting in the presence of subsurface and macro fractures of greater magnitude [31].

Wear is the process of gradual material loss on a solid surface caused by the movement of another substance in touch with it. Materials in contact experience relative motion in various applications. Occasionally, this movement is deliberate, such as in the rotation of plain bearings, the sliding of pistons in cylinders, the interaction between automotive braking disks and brake pads, or the manipulation of materials through machining, forging, or extrusion. Additionally, it could be inadvertent, such as in the case of fretting, which refers to minor cyclic movements that can lead to damage in specific structural connections when subjected to oscillating forces. Presence of solid particles, whether as impurities in a lubricant or purposely used in abrasive machining, will significantly impact the wear process.

Wear-resistant coating is a type of coating that serves a specific purpose. The wear-resistant filler can be uniformly and consistently dispersed on the surface of the coating film and is visible in a slightly raised state. The filler is the primary contributor to friction when the coating is exposed to friction. This protection of the coating not only prolongs its service life but also enhances its wear resistance. Wear-resistant coatings often obtain their wear-resistant properties by minimizing wear caused by external forces. This is accomplished through greater hardness and adhesion [32]. The performance of wear-resistant coatings is influenced by various factors, such as the composition of the base materials and fillers. Nanometer applications frequently employ wear-resistant coatings. The wear resistance of the coating can be significantly improved by including fillers that possess high strength, high hardness, and strong wear resistance [33]. Wear-resistant fillers in coatings can be classified into two primary categories: organic and inorganic. The organic fillers consist mostly of inert polymer materials such as polyvinyl chloride molecules and polyimide particles [34]. On the other hand, the inorganic fillers mostly contain SiC and metal flakes [35].

Certain applications require a lower level of slide resistance while placing a higher emphasis on its ease of cleaning. The higher the level of slip resistance of the floor, the

more challenging it will be to clean. This is because the aggregate that is applied to the surface of a highly slip-resistant component is both bigger and more angular in shape. Cleaning becomes increasingly challenging when the floor surface becomes more angular in its profile.

In general, it is preferable to enhance the slip resistance of floor surfaces. However, an excessively high coefficient of friction may hinder safe and comfortable walking. Moreover, a substantial augmentation in roughness poses challenges in the cleaning and upkeep of pavements, as it promotes the entrapment of debris within the uneven surface, thereby necessitating costly maintenance.

## 3. Research Progress on Anti-Slip and Highly Wear-Resistant Elastic Coatings

Since the 20th century, as the military power of the United States and the United Kingdom has grown, research on coatings that prevent skidding and resist wear has shifted from civilian use to military applications, particularly on decks. Deck anti-skid coating refers to the application of coatings that provide a non-slip effect on deck surfaces. The coating is primarily categorized into ordinary deck anti-skid coating and flight deck anti-skid coating. China's first comprehensive experimental ship, the "Liaoning" aircraft carrier is continuously improving and developing. This type of large military ship will have an expanding range of activities and will face increasingly challenging service environments, such as highly corrosive atmospheres and high temperatures. The combination of high pressure, pollution, and corrosion from chemical substances like seawater can lead to significant wear and corrosion damage. This necessitates strict performance and coating process criteria for deck anti-skid coatings.

Thus far, there have been advancements in the development of coatings that are both anti-skid and wear-resistant [36]. Various coatings can be chosen based on specific uses and requirements. The types exhibit a greater degree of diversification. The coating method can be categorized as spraying, brushing, and roller coating based on the technique used. The matrix can be broadly categorized as metal-based coatings and polymer-based coatings depending on its features. This article provides a concise overview of the research and implementation advancements in metal-based coatings and polymer-based coatings. It is based on the characteristics and classification of anti-skid and highly wear-resistant coatings. Additionally, it presents the future direction of development for anti-skid and highly wear-resistant coatings (Table 1).

**Table 1.** Comparison of Different Coatings.

| Type of Coating | Substrate | The Composition of a Coating | Anti-Slip Property (Friction Coefficient) | Wear Resistance [$\times 10^3$ kg mm$^{-1}$] | Intended Applications | Ref. |
|---|---|---|---|---|---|---|
| Metal-based | Metals, ceramics, polymers. | Alloy, amorphous, and metal matrix composite coatings layer. | ≥0.6 | 0.4~4 | Industrial production, offshore platforms, and ship decks, etc. | [4,7,18,20,29,37] |
| Epoxy-based | Previously painted surfaces, timber, natural and engineered stone materials. | Two-component anti-slip coating with epoxy resin as the main component. | ≥0.65 | 0.5~2.5 | Ordinary cars and passenger walkways, helicopter flight decks, ramps, etc. | [4,7,19,27,29,38–41] |
| Polyurethane-based | Concrete, steel, timber, stone surfaces, fiberglass. | Two-component coating with polyurethane resin as the main component. | ≥0.65 | 0.5~1.5 | Walkways for regular cars, machinery loading and unloading areas, etc. | [4,7,19,29,42,43] |

### 3.1. Metal-Based Anti-Slip and Wear-Resistant Coating

Metallic anti-skid coatings provide exceptional resistance to aging and decomposition, exhibit consistent friction coefficients, and demonstrate excellent anti-wear properties as well as high-temperature oxidation resistance. They provide extensive potential for use in the domain of anti-skid and highly durable coatings [44–47]. Extensive research has been conducted by both Chinese and international experts on anti-skid coatings made from metal. Currently, metal-based coatings can be classified into two primary categories: metal-based ceramic coatings and amorphous alloy coatings. Thermal spraying is a crucial technique used to prepare metal-based coatings that prevent skidding. This method includes many processes such as supersonic flame spraying, plasma spraying, arc spraying, and flame spraying.

In the early 1990s, the American NKF Engineering Company employed thermal spraying to apply a coating of aluminum wire infused with silicon carbide onto the steel deck surface of airplanes [48–51]. The field testing successfully demonstrated effective anti-skid properties. Prof. Petri Vuorist's research group at Tampere University of Technology [52] employed laser cladding to combine various metal and ceramic powders for the production of a metal-based ceramic covering. By carefully choosing metal matrix materials and carbide ceramics, most optimal wear outcomes were achieved.

Eungsun Byon et al. [53] from Korea Ocean University using twin-wire arc spraying (TWAS) technique to apply a recently created aluminum (Al) coating and an Al-3% titanium (Ti) coating onto a substrate made of high-strength low-alloy steel. This was accomplished by utilizing portable friction. The tester quantified the static and dynamic coefficients of friction of aluminum-based coatings under both wet and dry conditions. The findings indicate that the coating offers effective corrosion resistance for the steel base, significantly enhances the surface's friction coefficient, and exhibits favorable anti-skid characteristics. According to this study, the research group indicated earlier [54] has verified the practicality of using TWAS aluminum-based coating in the anti-slip section of marine structures. When compared to typical epoxy anti-slip coatings, TWAS aluminum-based coatings provide superior and consistent static friction coefficient values. Additionally, these coatings retain a certain level of strength even when subjected to isothermal exposure studies.

Furthermore, Chinese experts have conducted extensive investigations on anti-skid coatings that are based on metals. Prof. Li Hua's research group at the Ningbo Institute of Materials Technology and Engineering [55] employed flame spraying technology to fabricate a covering with anti-slip and wear-resistant properties, using a composite of aluminum and aluminum oxide (Al-$Al_2O_3$). When the concentration of $Al_2O_3$ was increased to 20 wt%, the study observed that the Al-$Al_2O_3$ composite coating exhibits superior anti-wear and corrosion resistance. During the salt spray corrosion process, the $Al_2O_3$ skeleton structure with dispersed fabrication acts as a barrier, preventing the entry of chloride ions into the coating's interior. This effectively decelerates the corrosion process, as depicted in Figure 2A. This coating exhibits significant potential for utilization in marine conditions, since it enhances and stabilizes the friction coefficient.

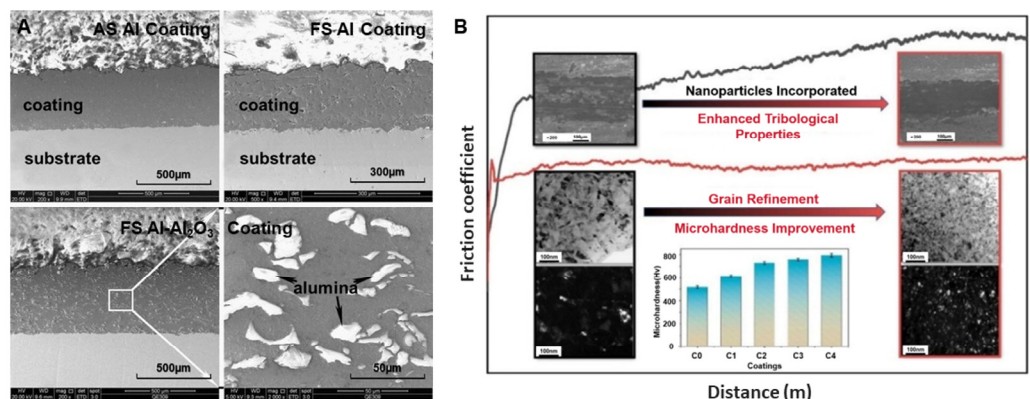

**Figure 2.** (**A**) SEM images of flame spraying (FS) and high-speed arc spraying (AS) Al and Al-$Al_2O_3$ coatings [56]; (**B**) comparison of coating morphology and performance before and after adding $Y_2O_3$ nanoparticles [37].

Prof. Li Yan's research group at China University of Petroleum [56] employed supersonic plasma spraying and cold spraying techniques to create the surface layer and middle layer of the anti-skid coating, respectively. This resulted in a sandwich structure that significantly enhanced the corrosion resistance of the NiCr-$Cr_3C_2$ anti-skid coating. Prof. Jiang's research group [57] used a nickel tungsten yttrium (Ni-W-$Y_2O_3$) composite coating on the surface of low carbon steel. An investigation was conducted to examine the impact of $Y_2O_3$ nanoparticles on the microstructure, morphology, microhardness, and tribological properties of the composite coating. This study is illustrated in Figure 2B: the

black line means the coating with 0 g/L $Y_2O_3$ content and the red line means the coating with $Y_2O_3$ content of 2 g/L. SEM images on the left are all without $Y_2O_3$ nanoparticles, while the ones on the right have $Y_2O_3$ nanoparticles. We can see relatively clearly that the findings indicate that the incorporation of $Y_2O_3$ nanoparticles has a notable impact on reducing the grain size of the coating, leading to enhanced mechanical and tribological characteristics of the coating. Although the addition of $Y_2O_3$ nanoparticles increased the anti-wear property, it reduced the anti-skid property.

Despite the numerous benefits of thermal spraying technique, its primary drawback is the incomplete density of the resulting coating. Extensive porosity has been found to greatly diminish the corrosion resistance of the coating [37,58]. Nevertheless, in numerous instances, the application setting and circumstances of anti-skid high wear-resistant coatings necessitate the coating to possess exceptional resistance against wear and corrosion [59,60]. Simultaneously, the upkeep of thermal spray coatings necessitates skilled sprayers, which hinders prompt repairs and incurs high expenses. Simultaneously, metal-based ceramic coatings encounter challenges such as excessive brittleness and inadequate impact resistance, necessitating the incorporation of tough materials to enhance their performance. In the case of amorphous coatings, the quality of the coating is significantly influenced by the level of amorphization. As it can result in an uneven distribution of coating materials, controlling the stability of the coating formed under certain process circumstances is challenging. Consequently, the preparation process becomes more complex. Metal-based anti-slip coatings are inferior to polymer-based coatings in terms of both anti-slip properties and high wear resistance due to these characteristics. Therefore, polymer-based coatings are more frequently employed in the domain.

### 3.2. Polymer-Based Anti-Slip and Wear-Resistant Coating

Resin-based materials, anti-skid granules, pigments, fillers, additives, and solvents are the primary components of polymer-based coatings that are designed to prevent skidding and withstand high levels of wear [61]. The resin base material, also known as the film-forming substance, is the primary constituent of anti-skid coatings. The film-forming resin component typically consists of resin or oil molecules containing specific functional groups. The performance of the base materials in the coatings controls the anti-slip and wear-resistant performances of coated substances. It has the ability to create a consistent and uninterrupted layer by combining various elements including anti-skid granules, pigments, and fillers, and applying it onto the material's surface.

Anti-skid granules serve as the primary component of anti-skid coating, primarily responsible for imparting anti-skid characteristics. Various shapes and sizes of anti-skid granules can be incorporated into the coating or applied onto the material's surface as required, resulting in a certain friction coefficient and providing anti-slip qualities. Pigments are auxiliary ingredients that contribute to the formation of a film and are mostly employed for the purpose of coloring paint. The pigments can be categorized into system pigments and coloring pigments. Coloring pigments are inorganic and organic pigments that give various colors to the application medium; system pigments are pigments that have no coloring or hiding power, and they were used in the early days to reduce the cost of products and to improve the performance of coatings, plastics, rubber, and so on.

Fillers are commonly referred to as fillers or extenders. Incorporating fillers can significantly enhance the performance of the paint coating, such as surface morphology and mechanical properties, and the most important are anti-slip and wear-resistant performances. Varying sizes of filler particles present varying challenges in achieving dispersion within the matrix resin, and their level of physical and chemical interaction with the matrix will also range. The use of various fillers in coatings leads to significant variations in viscosity, gloss, hiding power, and density. These variations have a direct impact on the hardness, adhesion, weather resistance, and media resistance of the paint layer. Additives are supplementary compounds that help form a film in coatings, and they are present in minute amounts. Appropriate utilization of additives can diminish coating flaws and enhance overall efficacy [62].

Currently, the prevailing anti-skid and wear-resistant coatings primarily consist of polymer-based coatings. These coatings offer the benefits of uncomplicated construction techniques, convenient operation, and satisfactory bonding strength, as well as meeting the anti-skid and wear-resistant requirements of most materials. Polymer-based coatings that prevent skidding and resist wear can be categorized into three types based on the substrate: epoxy resin-based coatings, polyurethane-based coatings, and other types of coatings.

### 3.2.1. Epoxy Anti-Slip and Wear-Resistant Coating

Due to its excellent anti-corrosion and chemical resistance properties, epoxy resin has emerged as the favored option for anti-skid and highly wear-resistant coatings. Epoxy resin coatings, formulated with precise proportions of specified constituents, exhibit exceptional characteristics such as high friction and wear resistance, impact resistance, and anti-skid capabilities. The utilization and advancement of anti-skid and very durable coatings has a long-standing history. In industrialized countries, epoxy anti-skid coatings were already being used on shipboard flight decks as early as the 1960s. Over time, several anti-skid coatings with exceptional performance were subsequently developed.

Through ongoing scientific study, the anti-skid and wear-resistant coating systems in Europe and the United States have seen significant improvements, resulting in their widespread recognition for their advanced anti-skid technology globally. The initial military standard for flight deck anti-skid coatings, MIL-D-23003 [63], was developed by the U.S. Department of Defense in 1961. The adoption of MIL-PRF-24667C (2008) [64] as the standard since 1961 [38] has prompted numerous firms to create novel anti-skid coatings. In 1983, researchers at the Naval Research Laboratory in Washington, D.C. [65] created a sturdier aircraft carrier deck. The durability of the paint is significantly enhanced. The paint is two component epoxy polyamides which can be cured at room temperature and use conventional solvents and pigmented with titanium dioxide, carbon black, and talc, which lasts 2 to 2.5 times longer than the current paint. The current paint contains low-molecular-weight aromatic naphtha and ethylene glycol mono-ethyl ether. Additionally, when the paint decomposes, it transforms into a powder form, thereby safeguarding the deck from any potential harm. Debris which destroyed pieces of the deck have the potential to be drawn into the engine, resulting in expensive damage.

In 1985, the American AAMC Company [66] developed an anti-skid coating with exceptional durability, specifically designed for ship decks. This coating demonstrated remarkable resilience to the impact and abrasion caused by numerous aircraft taking off and landing on the deck. The American AST Company [67] created the EPOXO 300C epoxy polyamide deck anti-slip coating in 1994. This coating incorporates granular high-hardness silica, which enhances its wear resistance. The findings indicated that the coating's friction coefficient remained nearly constant when exposed to both water and oil. Additionally, the coating exhibited excellent resistance to high temperatures, corrosion, and impact. Moreover, it displayed strong adherence to the underlying substrate. It is extensively utilized in the flight decks of United States warships and airplanes.

The Intershield 6 GV/6 LV and Intershield 9 L epoxy resin anti-skid coatings, created by the Netherlands-based International Paint Company (IP), exhibit a remarkably low rate of solar energy absorption and possess exceptional endurance. Furthermore, these coatings may be sprayed even under low-temperature conditions. The Proreco epoxy resin anti-skid coating, which originated in the UK, is extensively employed on ship decks and can also serve as a coating for helicopter decks [68]. The American DuPont Company [69] produced an epoxy anti-skid deck coating, MS-600, which consists of epoxy resin and Kevlar fiber. Subsequently, they performed friction and UV aging experiments on this coating. After undergoing a UV aging test for a duration of 2000 h, the coating exhibited no major alterations. The friction resistance rate was found to be below 10%, indicating exceptional resistance to aging, anti-slip properties, and wear resistance.

F. J. Friedersdorf et al. [70] invented and created an epoxy resin anti-skid coating that fulfills the specifications of the US military standard MIL-PRF-24667B [71]. Magnetic fillers were incorporated into the primer, conductive fillers were introduced into the intermediate

paint, and wear-resistant fillers as well as dielectric performance fillers were included in the topcoat. The findings demonstrated that the coating exhibited favorable anti-skid characteristics and robust impact resistance. Furthermore, it possesses wave-absorbing characteristics and a high coefficient of friction.

Chinese research on the development of anti-skid and highly wear-resistant coatings is limited. In the past, there was a lack of advanced military equipment like aircraft carriers, so early anti-skid coatings were primarily used on pavements in areas focused on traffic safety, such as sidewalks and driveways. The structure is seen in Figure 3A [72]: the majority of these materials consist of rubber elastomer, sand, and gravel, which are combined with a binder and applied on the road surface to enhance traction and prevent skidding. Subsequently, scientists innovated a novel composite anti-skid coating, mostly composed of resin infused with anti-skid particles. Simultaneously, the focus of research on anti-skid coatings was mostly on flat surfaces. The surface roughness of this coating is less pronounced, resulting in an improved look. It has a flat structure and is versatile, catering to various industries [73]. The Shanghai Kailin Paint Factory was the pioneer in China in terms of developing and manufacturing this particular variety of anti-skid paint. The researchers selected polysulfide rubber as the additive to enhance the epoxy resin coating and subsequently carried out performance evaluations on it. The findings indicated that the coating exhibits a significant elongation rate. The salt spray test may last for a maximum of 1500 h, and there is minimal alteration observed after immersing the coating in water for a period of 3 months. This suggests that the coating exhibits commendable flexibility and resistance to corrosion [74].

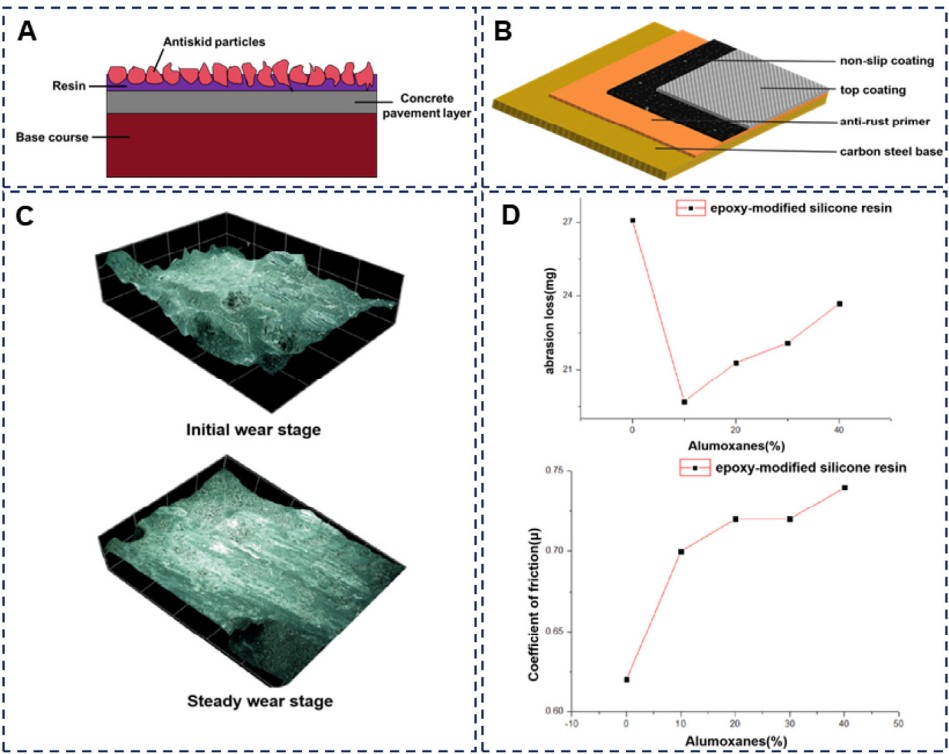

**Figure 3.** (**A**) Schematic diagram of the structure of the anti-skid coating on the road [72]; (**B**) schematic diagram of the structure of the anti-skid and highly wear-resistant deck coating; (**C**) three-dimensional morphology of the wear-marked surface of the anti-skid coating on the deck [39]; (**D**) Alumoxane effect on the wear resistance and slip resistance of resin [38].

Lin et al. [72] documented a patent for a durable anti-skid coating comprising a mixture of anti-skid paint slurry and a curing agent. The anti-skid paint slurry is formulated by combining polyester resin, titanium dioxide, lubricants, thickeners, additives, antioxidants, toughening agents, and epoxy resin. Effective anti-skid, anti-corrosion, and wear-resist

qualities can be attained through a combination of chemical and physical measures. Luo [75] documented the formulation of a solvent-based epoxy coating with several anti-skid additives. The coating has a maximum friction coefficient of 0.6.

Deng et al. [39] developed a novel deck anti-slip coating with a sandwich structure that exhibits exceptional resistance to wear. The coating primarily consists of a polyurethane modified epoxy anti-rust primer, an anti-skid layer, and a polyurethane finish. The arrangement is depicted in Figure 3B. The results indicate that the coating exhibits excellent friction performance and a high friction coefficient. As wear accumulates, the friction coefficient falls, but the rate of weight loss due to wear increases. As depicted in Figure 3C, the first stage is the initial stage of wear (cyclic wear 0 to 200 times); at this stage, contact wear occurs between the local raised portion of the coating surface and the abrasive parts, resulting in abrasive debris shedding. And the abrasive debris shedding scratches the surface of the coating under the combined effect of tangential pressure and normal pressure, resulting in a small number of scratches and shallow plough grooves appearing on the surface of the coating. The second stage is the wear stabilization stage (cyclic wear 200–500 times); with the increase in the number of wear times, the wear morphology of the coating surface becomes smoother, the wear process becomes smooth, and the wear loss rate increases slowly. The predominant wear mechanism is the early wear stage, followed by abrasive wear and adhesive wear during the steady wear stage.

This coating not only guarantees the safety of marine structures, but also establishes a theoretical foundation for the advancement and implementation of novel anti-skid and wear-resistant coatings for decks.

Liu and co-workers [38] employed Alumoxane particles to augment the SH-023-7 epoxy-modified silicone resin for the purpose of creating a non-slip covering. The findings indicate that the coating exhibits excellent resistance to high temperatures and wear. As depicted in Figure 3D, the optimal performance is achieved when the Alumoxane addition amount is 10 wt% of the resin base material. This results in a level 1 adhesion, a friction coefficient of 0.70, and a wear amount of 19.7 mg. Prof. Huang's research group [38] developed a water-based epoxy emulsified asphalt coating with excellent anti-skid and low-temperature characteristics. This coating can be utilized as a road maintenance material.

### 3.2.2. Polyurethane Anti-Slip and Wear-Resistant Coating

Polyurethane is a versatile polymer characterized by its ability to have a flexible and adaptable structural design. The molecular structure consists of a block copolymer that is made up of alternating segments with different properties: flexible segments (soft segments) and rigid segments (hard segments). The rod-shaped line segments indicate the hard segments of the macromolecular diisocyanate, and bead-string-like line segments are diisocyanate-coupled polyether soft segments, as depicted in Figure 4A. The supple section offers the durability and ability to deform under stress of the elastomer and, along with the rigid components, contribute to the stiffness and robustness of the material. The primary chain of polyurethane molecules contains not just urethane groups, but also polar groups such as ether, ester, or urea groups. The abundance of polar groups facilitates the establishment of many hydrogen bonds both within and between polyurethane molecules. Microphase separation readily occurs in linear polyurethane due to the thermodynamic incompatibility between its soft and hard segments. This allows for the formation of physical cross-links through hydrogen bonding [76,77].

Polyurethane elastomers possess notable structural attributes that confer onto them elevated strength, tear resistance, and exceptional wear resistance [78]. Polyurethane elastomers exhibit superior wear resistance compared to routinely utilized wear-resistant polymer materials, earning them the designation of "wear-resistant rubber". Germany achieved the full-scale industrial manufacture of polyurethane in the 1940s. Bayer O and other researchers pioneered the synthesis of polyurethane resin by utilizing the interaction between polyol and isocyanate [79]. Polyurethane exhibits high versatility, exceptional durability against wear and impact, and outstanding resistance to corrosion. Currently, polyurethane anti-skid coatings are extensively utilized as deck coatings in the whole world.

In 1972, A. Čížek et al. [80] in the United States created a polyurethane covering for decks that prevent skidding. In comparison to the anti-skid deck coating employed during that period, the coating exhibited greater thickness, resilience, and longevity that was durable against wear and forceful collisions. Cambon et al. [81] developed a polyurethane coating with multiple layers to create an anti-skid surface. This coating consists of a priming layer and one or more layers of anti-skid coating. The coating has excellent oil repellency and anti-slip characteristics, making it suitable for application on flight decks or offshore oil platforms which can be used for conducting exploratory activities.

Strait, J. S et al. [82] developed and documented a pliable polyurethane coating that serves as an anti-skid and wave-absorbing agent. This coating is capable of efficiently absorbing or dispersing radar signals in the microwave and millimeter wave range. Guo and his colleagues developed and documented a non-slip deck coating using an emulsion technique. The researchers opted for polyurethane or silicone-acrylic emulsion as the primary substance, then used polyurethane rubber particles as anti-skid fillers to produce the anti-skid coating. The findings demonstrated that the coating has excellent flexibility, adhesion, wear resistance, and resistance to various media.

Zhu et al. [83] developed and documented a polyurethane deck coating that effectively prevents slipping. This coating demonstrates excellent properties in terms of absorbing UV radiation, resisting environmental conditions, and withstanding the effects of aging. Coatings were prepared by Prof. Zhang's research group at the National Center for Nanoscience [84] by combining polyurethane–acrylate oligomers with varying amounts of nanosilica particles. The researchers conducted a comprehensive analysis of the structure and physical characteristics of the composite coatings. They also assessed the coatings' ability to resist wear and evaluated the behavior of the coated sample when subjected to reciprocating sliding, as shown in Figure 4B. The findings indicate that the inclusion of 40 wt% nanosilica particles leads to a 20% decrease in the friction coefficient of the coating. Additionally, the wear rate of the coating is approximately 70 times lower compared to the purified coating (the coating without nanosilica particles).

Chen et al. [85] conducted a study at the State Key Laboratory of Marine Coatings where they utilized modified ceramsite sand as anti-slip particles. They also changed the levels of polyol and isocyanate groups to create a lightweight, wear-resistant polyurethane covering with anti-slip capabilities. The study also examined the impact of particles on the creation of anti-skid coatings, the condition of the coating surface as depicted in Figure 4C, and the maximum static friction coefficient. The findings indicate that when the size of the anti-skid particles is between 250 and 380 μm and the amount added is 12%, the resulting coating exhibits reduced density, increased friction coefficient, enhanced wear resistance, and improved adhesion.

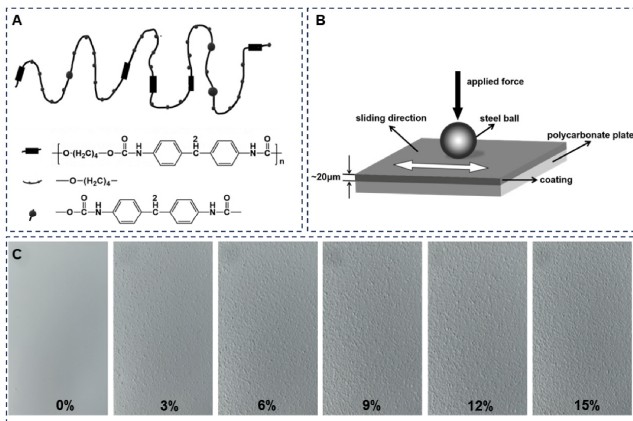

**Figure 4.** (**A**) Polyurethane molecular structure; (**B**) schematic diagram of a long-term fretting experiment [84]; (**C**) effect of the amount of anti-skid particles added on the morphology of the coating [85].

The presence of a significant quantity of organic solvents in polymer-based anti-skid coatings leads to environmental pollution during their production and application. Therefore, it is crucial to prioritize the development of environmentally benign polyurethane-based anti-skid coatings. In the 1970s, the German company Bayer pioneered the development and production of polyurethane curing agents that were free of solvents or had a high solid content. This polyurethane product can contain more than 90% solid content [86]. The AS2500 polyurethane anti-slip coating, created by American Safety Technologies (AST), is a solvent-free coating that contains 100% solid material. The non-volatilization of organic solvents [87] ensures that this coating does not have any negative effects on the environment.

Sun and his colleagues [88] created and formulated a polyurethane deck covering that lacks anti-skid particles. The coating possesses a distinctive molecular architecture and its surface is adorned with reactive functional groups. The test findings indicate that the coating exhibits a wear resistance of less than 0.018, satisfying the requirements of GB/T 1768. Furthermore, the surface remains free from any noticeable damage, blistering, or rusting after undergoing a 2000 h salt spray test. It has been effectively employed for a military deck and concrete trestle anti-corrosion project for a duration of 2 years, yielding favorable outcomes.

Liu and co-workers [42] utilized SiC fibers and alumina particles as composite anti-skid agents in order to fabricate reinforced polyurethane anti-skid coatings. The findings indicate that the coating exhibits optimal performance when the silicon carbide concentration is 9% and the alumina particle content is 15%. The abrasion rate is 1.1 mg, the adhesion strength is 6.38 MPa, and the non-volatile matter content is 74.1%, resulting in significant enhancement of the anti-slip characteristics. The coating exhibits both anti-skid and abrasion-resistant properties. Simultaneously, the coating exhibits robust adhesion, excellent flexibility, resistance to acid, alkali, and salt, and can undergo curing at ambient temperature. Liu and her colleagues at Baotashan Paint Company [43] developed a polyurethane coating with remarkable durability, superb anti-static characteristics, effective slip resistance, quick drying, high concentration of solids, and minimal environmental impact, which adheres to the regulations set by the national environmental protection authorities. Chao and colleagues [89] from the China University of Geosciences employed polyurethane and carbon nanotube particles to fabricate a polyurethane superhydrophobic coating. The findings demonstrated that the covering exhibited favorable characteristics in terms of preventing ice formation and enhancing traction.

### 3.2.3. Other Types of Anti-Slip and Wear-Resistant Coatings

In 1967, a silicone anti-skid coating was invented and manufactured by C. G. Cash et al. [90] of GE Company in the United States. This coating employs hydroxyl-terminated organosilicon compounds and incorporates diverse fillers. This material was anticipated to serve as a resilient, anti-skid coating for decks. In the late 20th century, H. Yoshioka et al. [91] developed an anti-slip coating by incorporating hydroxyl-containing silicone resin or silicone oil into organic resins like epoxy resin, polyurethane, and phenolic resin. They then evaluated the coating's static and dynamic friction coefficients, as well as other performance characteristics. The American TDA Company [69] has created and formulated an inorganic/organic hybrid resin that exhibits exceptional durability against wear. This resin not only enhances the flexibility and anti-slip characteristics of the coating, but also finds application in carrier-based aircraft and aircraft carrier decks.

Zhang [27] employed epoxy resin as a means to alter silicone resin. Figure 5A illustrates the reaction. The study involved the preparation of silicone anti-skid coatings and modified silicone anti-skid coatings by varying the amounts of curing agents, pigments, and functional fillers. The best ratio was determined and the anti-skid mechanisms of these coatings were investigated. The results indicate a significant improvement in the anti-slip and wear resistance properties of the resin-modified coating when compared to silicone resin. The addition of inorganic fillers enhances the anti-slip and wear resistance properties of the anti-slip coating. Prof. Li's research group at Tongji University [92] employed three different polymers, specifically acrylic resin, epoxy resin, and polyurethane resin, to

synthesize and create the surface system for anti-skid coatings. They conducted separate tests to measure the interface strength of each polymer. The findings indicate that acrylic resin glue anti-skid coatings exhibit inadequate thermal aging resistance, whereas epoxy resin and polyurethane resin glue anti-skid coatings provide comparatively superior thermal aging resistance. The interfacial strength of each polymer after water immersion and after oil corrosion varied differently. Under the action of water soaking and oil immersion, the interfacial strengths of the three resin antiskid systems were almost all reduced, but to a lesser extent. After UV irradiation, the interfacial strength of acrylic resin anti-slip system tends to decrease, and the interfacial strength of epoxy resin and polyurethane resin anti-slip systems showed an increasing trend.

The aforementioned study group [93] examined and investigated the impact of various composition materials on the friction coefficient and interface strength. The findings indicated that the friction coefficient of road pavement is mostly influenced by the stone's composition. Quartz sand is superior to colored sand and bauxite gravel in comparison. The friction coefficient is the highest and has superior anti-skid performance. Zhang Jun et al. [94] conducted a study at Air Force Engineering University where they developed coatings that were both anti-slip and wear-resistant. They achieved this by using carboxyl styrene–butadiene latex, water-based epoxy resin, and modified polyurea elastomer. Additionally, they investigated the effect of incorporating various granules on the friction properties of the coatings. This information can be found in Figure 5B. The findings demonstrate that incorporating cement and fiber into the coating can significantly improve its wear resistance. Notably, the polyurea elastomer exhibits the most pronounced enhancement effect, making it suitable for meeting the demands of anti-skid and high wear resistance on aircraft runway surfaces.

There is a considerable need for coatings that prevent skidding and withstand wear on asphalt pavements. Prof. Zheng's research team at Chang'an University collaborated with Qinghai Caojiabao Airport [40] to create a solar heat reflective coating specifically designed for asphalt pavement. This coating, depicted in Figure 5C, incorporates ceramic particles and machine sand to improve its anti-skid properties. The application of the coating is anticipated for utilization on airport pavements. Prof. Han's research group at Chang'an University [95–97] formulated a hydrophobic emulsified asphalt covering by incorporating a hydrophobic ingredient (HPA, polytetrafluoroethylene powder) into the emulsified asphalt. The results indicate that the coating has favorable hydrophobic, anti-icing, and anti-slip characteristics. Prof. Zhu's research team at Wuhan University of Technology [98] developed a colorful anti-skid coating to enhance the road surface's anti-skid performance. The composition of the coating was identified by experimental analysis: the anti-skid particles they used are emery, polyurethane particle, quartz sand, and color ceramic particle. Prof. Xu's research group [99] proposed a novel approach for achieving autonomous anti-icing on asphalt pavement. This method involves the use of an acrylic/polytetrafluoroethylene coating (APC). Studies demonstrate that the application of APC enhances the ability of the wear-resistant and transparent acrylate-based coating with highly filled nanosilica particles of asphalt pavement to resist icing, while also exhibiting exceptional characteristics in terms of slip resistance and water resistance. Prof. Jiang's research group [41] conducted a study on the variability of temperature resistance, water stability, wear resistance, and anti-skid capabilities of six different types of colored heat-reflective asphalt coatings. The results indicated that, among the coatings of varying hues, the red coating exhibited the most superior cooling performance. Simultaneously, the incorporation of anti-skid particles enhances the material's anti-skid capabilities while maintaining its cooling performance unaffected. M. Pomoni et al. [100,101] at National Technical University of Athens in Greece aimed to investigate the anti-skid of asphalt mixtures containing two different types of recycled materials, such as hot-mix asphalt (HMA) and mixtures containing either reclaimed asphalt pavement (RAP) for aggregate replacement or crumb rubber (CR) as a bitumen additive. They used British Pendulum Tester (BPT) test anti-skid performance at different temperatures and different surface

conditions. The result showed that the two types of coatings both showed great anti-skid properties. Overall, it has great potential for use in asphalt anti-skid pavements.

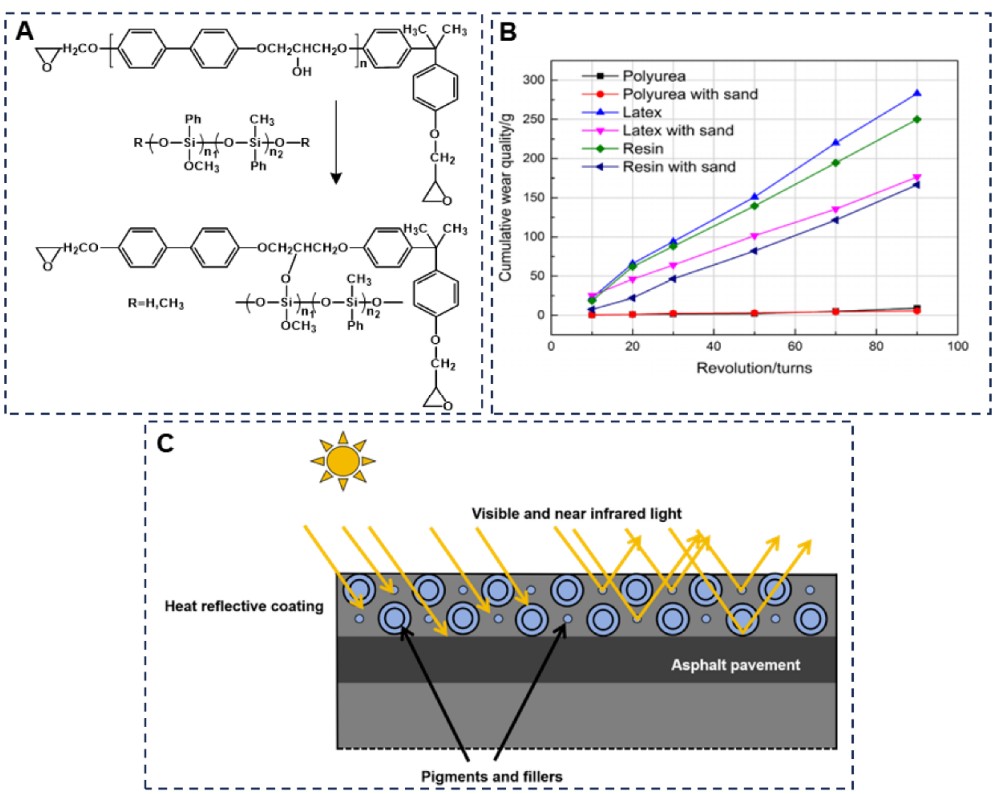

**Figure 5.** (**A**) Mechanism diagram of the modification of silicone by epoxy resin [27]; (**B**) changes in cumulative wear mass of different surface samples with the number of revolutions [94]; (**C**) working principle of solar reflective coating [40].

To summarize, while polymer-based coatings are presently the predominant choice for anti-skid and extremely wear-resistant coatings, they nevertheless possess certain inherent deficiencies. Being an organic covering, it inevitably exhibits drawbacks such as limited resilience to high and low temperatures, susceptibility to aging and breakdown, and a certain level of environmental impact. For many years, there has been a persistent issue that needs to be addressed by anti-skid and wear-resistant coatings.

## 4. Summary and Prospects of Anti-Slip and High Wear-Resistant Coatings

This study examined the accomplishments in anti-skid and wear-resistant coatings, focusing on the mechanisms behind their anti-slip and wear-resistant properties, the intrinsic qualities of the materials, and their extensive applications. Metal-based or polymer-based coatings can successfully address the requirements for anti-skid and wear resistance in many application scenarios. Simultaneously, incorporating various anti-skid particles enhances the friction coefficient between the material and the substrate, hence boosting its anti-skid performance to a certain degree. Specific surface texture can enhance both anti-skid and wear-resistant qualities to a considerable extent.

The issue of adapting to different surfaces is a pressing matter that requires a solution. Polymer-based coatings that provide both anti-slip and wear-resistant properties have been widely employed in various applications, including decks and pavements. Their basic materials are readily available, their production processes are well-established, they exhibit excellent bonding capability with the substrate, and they are highly durable. Nevertheless, there exists a considerable array of organic solvents that are harmful to the environment and human health when utilized in the formulation and production of polymer-based coatings. Hence, the main focus of future research lies in the development of novel eco-friendly

coatings that possess both anti-slip and wear-resistant properties, while also enhancing their longevity.

Future study is primarily focused on the fundamentals and practical uses of cutting-edge materials that prevent skidding and resist wear. Optimizing the setup and post-processing process parameters can effectively enhance the anti-skid and high wear-resistant qualities of materials through large-scale refinement of nanostructures.

**Funding:** This research received no external funding.

**Institutional Review Board Statement:** Not applicable.

**Informed Consent Statement:** Not applicable.

**Data Availability Statement:** Data are contained within the article.

**Conflicts of Interest:** All authors were employed by the company Wuhan Research Institute of Materials Protection.

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
