# Peer review of "Recent Progress on Anti-Slip and Highly Wear-Resistant Elastic Coatings: An Overview"

_coatings, doi:10.3390/coatings14010047_

Round 1

Reviewer 1 Report

Comments and Suggestions for Authors

1. In the introduction, the author should define the structure of the review. Further, add a few papers from 2023 that strengthen the review in terms of the latest developments.

2. The author should also include the techniques used in anti-slip coating.

3. The author should also include the techniques used to investigate the friction and wear before wear mechanism.

4. The author should use the tabular form of data for better clarity in the different materials-based anti-slip coating. Further, make a comparative table, which is best.

5. Challenges of anti-slip coating should added before summary.

Reviewer 2 Report

Comments and Suggestions for Authors

In this paper, this article reviews the design, performance and application of anti-skid
and high-wear-resistant coating materials at home and abroad. First, it introduces the structure and mechanism of anti-skid and wear-resistant coatings. The application mostly encompasses airplane and ocean decks, as well as pedestrian spaces. The review introduces the development status and research progress of metal-based anti-skid coatings and polymer-based anti-skid coatings, which are two groups of pavement. Finally, the challenges and future development directions of this key field are summarized and prospected.

The paper could be considered for publication in the Journal after the following major revisions:

1-use more concise keywords.   

2-Define in the abstract what aspects of anti-skid coatings were investigated, before briefly mentioning the results of such tests. Perhaps mentioning a few materials used would be useful.

3-Check the English of the whole paper.

4-move this sentence to introduction:

There has been great interest in research and development of different anti-skid and highly wear-resistant materials that can effectively reduce energy losses and improve efficiency in numerous applications.”

5-Introduciton should be strengthened. To modify this section the following documents can be consulted:

 doi: https://doi.org/10.1016/j.surfcoat.2023.129620 ,  doi: https://doi.org/10.1016/j.jmrt.2023.08.215

6-expalin anti-slip and highly wear resistance coating materials first before talking about the mechanism. Perhaps a table would be useful.  

7-figure 1a doesn’t have a scale bar.

8-figure 2 is hard to comprehend. A better quality figure is needed.

9-explain the features in figure 4.

10-consult the following references in the discussion section:

doi: https://doi.org/10.1016/j.ymssp.2023.110117  ,  doi: https://doi.org/10.1016/j.apsusc.2022.153982

11-better figure captions should be used. More concise and to the point

12-explain the drop in figure 3d (top curve).

13-any image for “Polymer-based anti-slip and wear-resistant coating”

14-conclusions should be more concise. They are somehow wordy. Make them quantitative as well.

15-better describe the rationale of the work at the end of the introduction.

Comments on the Quality of English Language

3-Check the English of the whole paper.

Reviewer 3 Report

Comments and Suggestions for Authors

This is an interesting paper with suitable content for this journal. The following comments are recommended to the authors:

1_ Please justify why the selected applications are primarily for airplane and ocean decks.

2_ Please include an objective statement, which is necessary even for a review paper.

3_ More information is also needed about investigations on the skid resistance in roadway pavements. A multitude of materials exist including non-conventional, recycled materials that could exhibit a sustainable potential for inclusion in airplane and ocean decks too. Please consult and probably cite other relevant studies from roadways, e.g., https://doi.org/10.3390/vehicles2010004 , https://doi.org/10.3390/recycling7040047 , etc.

4_ In chapter 4, study limitations, research gaps and open challenges as future prospects should definitively be discussed. Please revise.

5_ Please try to include more recent references, from year 2021 and so on to ensure the timely nature and the originality of the topic.

Comments on the Quality of English Language

Moderate changes are needed.

Reviewer 4 Report

Comments and Suggestions for Authors

Chen et al. briefly reviewed many studies on anti-slip and wear-resistant coatings with 83 references. The authors explained basic ideas regarding how coatings can provide coated substrates with anti-slip and wear-resistant properties in Section 2. Then, they categorized recently reported anti-slip and wear-resistant coatings into metal-based and polymer-based coatings in Section 3 and outlined some properties of these coatings. As for polymer-based coatings, they discussed them in three different subcategories: epoxy, polyurethane, and others. In the last section, they briefly summarized this field and provided an outlook.

Overall, the contents seem to be fine and potentially attract readers in this field and those from different fields but with interests in coating technologies.

However, at this stage, I recommend revising the manuscript so that it can provide information more precisely and informatively. The serious issues that the authors need to address are:

[a. Serious Issues]

(a–1) Almost no information about the presented figures

I feel this manuscript does not use figures to help readers understand the discussed points effectively because, for most of the figures, the authors did not provide sufficient information about what they are. Such information should be written in either the main text or the figure caption (I think describing it in the figure caption is helpful as readers do not need to look for it in a relevant section in the main text). For instance, in Figure 1A, readers will not understand what the authors would like to say using four subfigures. The appearances of these subfigures are different from each other, but no information is provided to explain why they are different (e.g., because of the difference in the coating procedure, particle size, and/or particle shape). Reading the cited references would help readers understand the discussed points, but always expecting this might be very inconvenient for readers as it is time-consuming and, sometimes, it might not be possible because of the limited readers’ access to the cited references. For the other figures, I will try to clarify what is missing in my major comments, but could the authors please check the figures and provide ample information in the manuscript to help readers understand the discussed points?

(a–2) Almost no comparison in the anti-slip and wear-resistant performances between different coatings

In Section 3, it would be difficult for readers (especially those who are from different fields) to understand the significance of each coating, which is because the authors’ narrative is like this: “Person A did X, person B did Y, and person C did Z, …(this continues towards the end).”, almost no correlation between past studies (and/or no improvement in a newer study from an older study) is stated, and most of the anti-slip and wear-resistant performances of discussed coatings are unquantified. This means that the manuscript appears to just collect points, but it does not provide sufficient correlation between these points.

I suggest that the authors summarize all coatings (discussed in the manuscript) in a table format and tabulate the quantified anti-slip and wear-resistant performances. In the table, possible columns are: Type of coating (i.e., metal-based, epoxy-based, polyurethane-based, or others), Substrate, The composition of a coating, Coating thickness, Anti-slip property (e.g., friction coefficient), Wear-resistant property (e.g., wear amount, wear resistance), and Intended applications (e.g., decks, roads). This allows readers to compare the different coating technologies clearly and identify some promising coatings for different applications.

The authors are recommended to address the above issues first as well as the other reviewers’ comments to improve the readability and significance of the manuscript.

Although resolving the above issues is the top priority, the following point-by-point suggestions would help improve the manuscript as well:

[b. Major comments]

(b–1) Lines 46, 137, 163, 287, 358, 359: Coatings is an international journal and readers reside across the globe. Hence, I feel using “nation”/“domestic” to refer to “China”/“Chinese” should be avoided (as readers would not be living in China and, in this case, their nations are other countries). The authors need to check throughout the manuscript and rephrase it.

(b–2) Figure 1A and Lines 73−75: As I stated in (a–1), readers will not understand Figure 1A based on the provided information. Please explain each subfigure in Figure 1A.

(b–3) Figure 1B and Lines 75−78: Similarly, Figure 1B is confusing. Which subfigures are the image before/after the addition of nanoparticles and before/after the wear test? What are the conditions for the wear test? Please specify them in the figure caption (or in the main text).

(b–4) Line 91:The use’s text is “[19];”.” can be deleted unless the authors would like to state something here clearly.

(b–5) Lines 98−99: I feel “Wear-resistant coatings often exhibit a low coefficient of friction” contradicts what the authors stated for anti-slip coatings (i.e., “An anti-slip coating is a type of coating that is applied to materials in order to enhance the level of friction on their surfaces” in Lines 61−62). Is it true? If it is true, how wear-resistant coatings can improve the friction coefficient? What are the wear-resistant coatings that cannot improve the friction coefficient? Please elaborate on them in the main text.

(b–6) Line 170: I could not understand “with dispersed strengthening properties acts as a barrier” because “properties” are not physical substances that can be dispersed in the coating layer. Please rephrase it to explain the context in more detail.

(b–7) Figure 2A and Lines 171−172: About (a–1), please specify differences between four subfigures in Figure 2A. Plus, I recommend remaking the scale bars for these subfigures as they are almost illegible.

(b–8) Figure 2B and Lines 182−185: Again, in relation to (a–1), please explain all the insets in Figure 2B. What are the black and red lines, x-axis, and y-axis in the biggest subfigure in Figure 2B? The labels and scale bars for the insets are illegible, so please enlarge Figure 2B so that readers can understand this figure precisely.

(b–9) Lines 200−202: This sentence needs revision because it is confusing in terms of what “it” refers to. If “it” refers to “amorphous coatings” or “amorphization”, swapping the first and second clauses will improve the readability (i.e., “As it can result in an uneven distribution of coating materials, controlling the stability of the coating formed under certain process circumstances is challenging.”). Alternatively, the authors can replace “it” with the full noun that “it” refers to.

(b–10) Lines 212−213: I feel this sentence “The performance of the coating has a significant impact on its overall performance.” does not make sense because it is an already known thing: i.e., The coating controls the anti-slip and wear-resistant performances of coated substances. In addition, the authors need to specify what “performance” refers to (although I guess it would be the anti-slip and wear-resistant performances). Please check the sentence and revise/delete it so that it matches the flow of this paragraph.

(b–11) Line 223: Similarly, what does “performance” refer to?

(b–12) Lines 255−256: The authors need to specify (the compositions of) the two paints compared here: i.e., “the paint” in Line 255 and “the current paint” in Line 256. At this rate, readers will not understand what they are.

(b–13) Line 257: What does “Debris” refer to? Is it “a powder form” of the paint or referring to much larger objects like the destroyed pieces of the deck or something? Please specify it.

(b–14) Figure 3A: What is the gray layer between “Resin” and the bottom “Base course”/”Concrete pavement layer”?

(b–15) Figure 3C and Lines 318−320: Concerning (a–1), please provide more information about the wear mechanism in this sentence. I could not understand the wear mechanism from this sentence and Figure 3C. Why do the authors think “The predominant wear mechanism is the early wear stage” and “abrasive wear and adhesive wear” damaged the surface “during the steady wear stage”? Based on Figure 3C, I can just know the 3D surfaces. In addition, please specify the conditions of the wear test.

(b–16) Figure 3C: Please provide subfigures with the scale bars.

(b–17) Figure 4A and Lines 348−349: The sentence is not supported by Figure 4A where no cross-links are depicted. Please check the sentence and Figure 4A and revise one or both of them.

(b–18) Line 362: Why is “thickness” important? Plus, why did “the anti-skid deck coating employed during that period” not provide a thick coating?

(b–19) Lines 384−385 (Figure 4B): The sentence needs a revision to specify Figure 4B is the device that evaluates “the behavior of the coated sample when subjected to reciprocating sliding” discussed in the preceding sentence.

(b–20) Line 387: What is “the purified coating”?

(b–21) Lines 393−394 (Figure 4C): From Figure 4C, readers will not understand the authors of this work “investigated the wear characteristics of the coating during various phases of wear”. Please revise the sentence.

(b–22) Line 448 (Figure 5A): I think “mechanism” can be deleted because Figure 5A does not show “the reaction mechanism”.

(b–23) Lines 460−461: I feel “The interfaces between water immersion and oil corrosion will be affected to different extents.” does not fit into the context discussed before/after this sentence. Could the authors please elaborate on this?

(b–24) Line 495−496: Why did “the red coating” exhibit “the most superior cooling performance”?

(b–25) Lines 496: The authors are encouraged to explain what “cooling performance” means.

(b–26) Line 504−505: The authors need to explain what “certain inherent deficiencies” are in more detail. This is important for readers to understand the ongoing challenges in polymer-based coatings.

(b–27) Lines 525: What does “efficiency” mean? How is it quantified?

(b–28) Line 531−533: This sentence appears not to fit into the context. I think the authors need to say that the current polymer-based coatings use organic solvents that are harmful to the environment, so the use of eco-friendly solvents and the development of solvent-free coating techniques are the focal points of future research.

[c. Minor comments]

(c–1) Lines 46−54: Please cite relevant references to support the descriptions in this paragraph.

(c–2) Lines 126−129 and 138: I think “anti-skid” can be deleted because these paragraphs state the coatings with both anti-skid and anti-wear properties. Ideally, “metal-based anti-skid coatings” are stated as “metal-based anti-skid and anti-wear coatings”, but if it is lengthy, simplifying the description is an option. Please double-check the other parts in the manuscript as well.

(c–3) Lines 173: I think “it surpasses pure aluminum coatings in enhancing” can be simplified to “it enhances” because I do not understand the necessity of discussing the reduction in the aluminum content based on the provided information.

(c–4) Lines 219−222: I think these two sentences can be revised to provide more precise information. Based on them, I understand “system pigments” are the pigments that “contribute to the formation of a film” and “coloring pigments” are the pigments that are “employed for the purpose of coloring paint”. Is it correct? If so, it would be nice if the authors could specify them in the main text.

(c–5) Line 282: Why were “conductive fillers” used?

(c–6) Line 415: It would be nice if the authors could briefly explain the definition of a quantified “wear resistance” here (i.e., what a wear resistance of 0.018 means).

(c–7) Line 439: The author’s name “H. Yoshionka” should be “H. Yoshioka” (please remove “n”).

(c–8) Line 487: It would be nice if the authors could specify the composition of the coating as they say “The composition of the coating was identified by experimental analysis”.

Comments on the Quality of English Language

The quality of the English language in this manuscript appears to be OK, but there are some errors and concerns as follows:

(1) Line 168: I think “The” between “composite” and “coating” can be deleted.

(2) Lines 204205: The phrase “Frequently employed in the domain.” has no noun and appears to be an incomplete sentence. Please check it.

(3) Line 213: Here, “Efficiency.” is an incomplete sentence. Please revise it.

(4) Lines 309310: The phrase “Properties that prevent corrosion and resist wear.” has no verb and appears to be an incomplete sentence. Please check it.

(5) Lines 325−326: The phrase “, as depicted in the figure” can be deleted as the same phrase appears immediately after this sentence.

(6) Line 361: The sentence “They enhanced it.” might be redundant.

(7) Line 367: Here, “Platform for conducting exploratory activities.” is an incomplete sentence. Please revise it.

(8) Line 409: I think “volatilization” should be replaced with “non-volatilization” because the preceding sentence discusses the coating technology without using solvents (i.e., no volatilization of solvents occurs in this case).

(9) Lines 422423: Please insert a space between a number and a unit (i.e., “1.1mg” should be “1.1 mg”).

(10) Lines 430−431: Here, “Adhere to the regulations set by the national environmental protection authorities.” is an incomplete sentence. Please revise it.

Round 2

Reviewer 1 Report

Comments and Suggestions for Authors

Thank you for incorporating the suggestions/comments.

Reviewer 2 Report

Comments and Suggestions for Authors

The revised version of the paper can now be accepted for publication. 

Reviewer 3 Report

Comments and Suggestions for Authors

None.

Comments on the Quality of English Language

minor checks are needed

Reviewer 4 Report

Comments and Suggestions for Authors

The authors have revised the manuscript as per the reviewers’ suggestions. Now the manuscript has been improved significantly. However, the authors are recommended to address the following minor comments to revise it before its acceptance for publication.

(1) Table 1: It would be nice if the authors could consider adding one column on the right side and add references for each “Type of coating”. The authors do not need to add new references for this, but if they can tabulate the existing references into this table, readers can easily find the relevant references for each coating category without checking the later sections, thereby improving the informativity of the manuscript.

(2) Figure 2A, Lines 231−232: The caption should include the abbreviations: flame spraying (FS) and high-speed arc spraying (AS).

(3) Figure 2B, Lines 232−233: Based on the friction coefficient vs. distance curves, the friction coefficient of the coating decreased when Y2O3 nanoparticles were added. Does this mean the addition of Y2O3 nanoparticles reduced the anti-skid property? If so, this adverse effect needs to be stated in the main text. (e.g., Although the addition of Y2O3 nanoparticles increased the anti-wear property, it reduced the anti-skid property.)

(4) Line 307: Concerning my previous comment (b−12), the authors need to specify the composition of “the current paint” as follows: “the paint containing low-molecular-weight aromatic naphtha and ethylene glycol mono-ethyl ether”.

(5) Line 449: In relation to my previous comment (b−20), the authors need to specify the composition of “the purified coating” as follows: “the coating without nanosilica particles”.

(6) Line 454: The position of “as depicted in Figure 4(C)” needs to be after “…the condition of the coating surface” (without a comma between them).

(7) Line 519: About my previous comment (b−23), the authors need to explain “the influence of application conditions on the interfacial strength of anti-skid surface system was studied using water soaking, oil immersion and UV irradiation methods” in the main text.

(8) Lines 522−523: In relation to the above comment, this sentence needs to be revised to avoid misunderstanding (=the focal point is the interfaces between water and oil by reading this sentence). For example, “The interfacial strength of each polymer after water immersion and that after oil corrosion varied differently.

(9) Line 559: This comment is just out of curiosity related to my previous comment (b−24). Why did different colors cause different cooling performances? Was it related to the difference between the color properties such as brightness and wavelength, or was it related to the difference between the chemical compositions of coloring pigments?

In addition, why did the six 0 kg/m2 coatings show different temperature profiles? I assume they did not contain any coloring pigments, i.e., their compositions were the same as each other. In this case, all 0 kg/m2 coatings should show the same temperature profile.

Comments on the Quality of English Language

Please correct the minor errors:

(1) Line 256: The verb “control” should be “controls”.

(2) Line 304: I suggest separating the clauses for better readability. Please consider replacing “,” with “.” to separate them.

(3) Line 304: “…the paint are” should be “…the paint is”.

(4) Line 307: I believe “lasting” should be used in the verb form, i.e., “lasts”.
